# Telemonitoring of Real-World Health Data in Cardiology: A Systematic Review

**DOI:** 10.3390/ijerph18179070

**Published:** 2021-08-27

**Authors:** Benjamin Kinast, Matthias Lutz, Björn Schreiweis

**Affiliations:** 1Institute of Medical Informatics and Statistics, Kiel University and University Hospital Schleswig-Holstein, 24105 Kiel, Germany; Bjoern.Schreiweis@uksh.de; 2Department of Cardiology and Angiology, University Hospital Schleswig-Holstein, 24105 Kiel, Germany; Matthias.Lutz@uksh.de

**Keywords:** telemonitoring, telemedicine, telecardiology, cardiology, wearable, sensors, consumer health devices, cardiovascular disease, heart failure, atrial fibrillation

## Abstract

Background: New sensor technologies in wearables and other consumer health devices open up promising opportunities to collect real-world data. As cardiovascular diseases remain the number one reason for disease and mortality worldwide, cardiology offers potent monitoring use cases with patients in their out-of-hospital daily routines. Therefore, the aim of this systematic review is to investigate the status quo of studies monitoring patients with cardiovascular risks and patients suffering from cardiovascular diseases in a telemedical setting using not only a smartphone-based app, but also consumer health devices such as wearables and other sensor-based devices. Methods: A literature search was conducted across five databases, and the results were examined according to the study protocols, technical approaches, and qualitative and quantitative parameters measured. Results: Out of 166 articles, 8 studies were included in this systematic review; these cover interventional and observational monitoring approaches in the area of cardiovascular diseases, heart failure, and atrial fibrillation using various app, wearable, and health device combinations. Conclusions: Depending on the researcher’s motivation, a fusion of apps, patient-reported outcome measures, and non-invasive sensors can be orchestrated in a meaningful way, adding major contributions to monitoring concepts for both individual patients and larger cohorts.

## 1. Introduction

Within the last decade, advances in sensor technology have made a large number of wearables and other consumer health devices ready for the market. Both leading technology companies and specialized manufacturers have acknowledged a need for affordable and accessible integrated sensor technologies for fitness and health; they are serving this trend with significant investments in the emerging market [1]. One result is a progressive penetration of these technologies into a large proportion of the general public, given that consumer health devices allow individuals to measure cardiac vital signs while working out, or to self-monitor their own health status, potentially improving an individual’s health behavior [2]. As these technologies become more widespread and sophisticated, there are many potential applications and use cases; several of these involve monitoring individual patients’ and entire cohorts’ physiology in the context of everyday life. This potential has been recognized by both researchers and health care professionals, as remote patient monitoring opens up new sustainable ways to support and care for patients in their homes [3,4,5]. In particular, the field of cardiology can be considered one of the most important fields of application, as integrated sensor technologies allow a variety of use cases, following up with a patient’s cardiovascular health status under real-world conditions while avoiding clinical biases such as white coat hypertension [6,7,8]. On the other hand, cardiovascular diseases are the leading cause of death in the European countries and, therefore, avoiding these has a huge impact on public health and the health system. For example, heart failure affects approximately 26 million people worldwide [9]. Once hospitalized, up to 25% of heart failure patients are readmitted within 30 days [10,11]. Thus, recognizing the worsening of heart failure and avoiding hospital admissions is a key quality metric for managing heart failure patients.

This also influenced the researchers of the Use Case Cardiology (UCC) of the HiGHmed [12] consortium when planning the integration of both institutional and cross-sectional heart failure (HF)-related health care data in 2017. As part of an affiliated telemonitoring (TM) study, the application of wearables in the follow-up care of HF patients is planned. The aim is to support patients and their physicians in the disease management of HF while simultaneously aggregating health data from the “black box” home setting by equipping patients with wearables, complementary devices, and patient-reported outcome measures (PROMs). The aggregated data will then be transferred into a medical data integration center and merged with the hospitals’ electronic health records (EHRs) to create a longitudinal dataset of HF patients. Therefore, it is the consortium’s premise to develop and deploy low-threshold state-of-the-art solutions. In doing so, our aim is to passively observe the patients’ disease progression retrospectively, without requiring any additional intervention. Thus, our study focuses on the latest consumer technologies that are suitable for everyday use.

In order to obtain an overview of recent research- and technology-related developments in the field, the main objective of this systematic review is to investigate the status quo of studies monitoring patients with cardiovascular risks and patients suffering from cardiovascular diseases in a telemedical setting, using not only a smartphone-based app, but also consumer health devices such as wearables and other sensor-based devices. With this novel approach, we aim to provide a holistic perspective on telemonitoring as we take both the general organizational and technical context as well as qualitative and quantitative aspects into consideration.

## 2. Materials and Methods

We performed a systematic review in order to identify published articles regarding telecardiological studies using consumer health devices to monitor patient’s’ health status reported via a mobile app. We identified and evaluated the available literature in accordance with the Preferred Reporting Items for Systematic Reviews and Meta-Analyses (PRISMA) guidelines [13] though no registration of the protocol was performed.

### 2.1. Search Strategy

We conducted this comprehensive and systematic search of five databases for literature published between 1 January 2001 and 31 March 2021. We identified relevant English-language publications searching *PubMed*, *Web of Science*, *CINAHL*, *Cochrane Library*, and *Scopus*. The mandatory keywords (“telemedicine” OR (“telecardiology” AND “cardiology” OR “cardiovascular disease”) AND “app” OR “mobile application”) were used for the search. We provide detailed queries in Table 1.

### 2.2. Inclusion and Exclusion Criteria

We intended to include articles matching the following criteria: (1) primary studies dealing with (2) telemedical concepts in (3) cardiovascular disease monitoring that used (4) consumer health devices such as wearables (5) or other noninvasive sensors to (6) track patients’ health data (7) with a smartphone app as a central user interface. Studies not considering both wearable- and sensor-generated data were excluded.

### 2.3. Selection and Data Extraction

We managed the retrieved articles of each search in the aforementioned databases with Citavi 5 (Swiss Academic Software GmbH Citavi 5 Version 5.7.1.0., Wädenswil, Switzerland). First, we removed duplicates; then, we identified relevant articles by screening all keywords, titles, and abstracts based on our selection criteria. We excluded all records that did not clearly meet the eligibility criteria. Subsequently, one experienced expert in the field of medical informatics assessed all potentially relevant and freely available full-text publications regarding the inclusion and exclusion criteria. In case of ambiguity, the articles were discussed with a second expert in the domain to decide about inclusion or exclusion. While we conducted the full-text review, we identified potentially relevant references in the first-level results based on the context.

### 2.4. Comparison Criteria

In order to compare the studies, we determined various comparison criteria and divided them into the three groups: (1) study protocol, (2) technical parameters, and (3) qualitative and quantitative parameters.

#### 2.4.1. Study Protocol

This group includes the framework conditions of the publications, giving an overview of the relevant studies. As this review focuses on cardiological diseases, the disease-related use cases form an important criterion together with both sample sizes and study cohort sizes, study types, and the minimum participation duration. In addition, the country in which the study was conducted, as well as the application area distinguishing between local (e.g., Munich), regional (e.g., Bavaria), or national (e.g., Germany), were selected.

#### 2.4.2. Technical Parameters

Consisting of whether study staff monitored patients by including intervention—i.e., actively intervening by adjusting a participant’s treatment/therapy plan (e.g., due to changing measures or vital signs)—or without intervention—i.e., as a passive, observing character. Additionally, the platforms on which the patients’ apps were offered were included. The third and fourth technical criteria include the applied wearables and other non-wearable consumer health devices connected to the patients’ apps.

#### 2.4.3. Qualitative and Quantitative Parameters

Following the group of technical parameters, this group focuses on parameters provided by (1) the patients, (2) measured via a wearable or other consumer health device, and (3) data collected in a hospital setting by a physician, including examinations and surveys. We further divided patient-reported data into patient-reported outcome measures (PROMs) and patient-generated health data (PGHD). PROMs, following the definition of Weldring et al., describe tools or instruments (e.g., standardized questionnaires) developed to ensure a valid and reliable measurement of patient-reported outcomes [14]. Accordingly, these can further be subdivided into the PROMs that (a) measure functional status, or capture (b) health-related quality of life, (c) symptoms and symptom burden, (d) personal experience of care, and (e) health-related behaviors such as anxiety and depression, as well as PROMs that cannot be assigned to any of the above-mentioned groups because they are, for example, non-disjunct, summarized into (f) others. While, according to Sharpio et al., patient-generated health data (PGHD) are defined as ”health-related data including health history, symptoms, biometric data, treatment history, lifestyle choices, and other information created, recorded, gathered or inferred by or from a patient”‚ in this review we focus on patient data documented via an app [15]. Finally, the specific vital signs provided by wearables and other consumer health devices were also included as a criterion, while the frequency by which device-tracked parameters were captured was also taken into account.

## 3. Results

We identified 166 articles in our initial search (see Appendix A: Databases export). After we removed duplicates, a total of 157 articles were included for the title and abstract screening process. Among these, 30 articles seemed relevant, and we performed a full-text review/evaluation, resulting in a total of 7 articles being eligible and included in the study (see Appendix A: Articles excluded) [16,17,18,19,20,21,22]. After we did a backward reference screening, we included one additional article [23]. Finally, eight articles were included in this systematic review. The detailed selection process is illustrated as a PRISMA flow diagram in Figure 1.

The included articles describe studies with several different types of study design, such as proof-of-concept studies (2 of 8; 25.0%) [19,22], randomized controlled trials (2 of 8; 25.0%) [20,23], cluster randomized trials (1 of 8; 12.5%), longitudinal cohort studies (1 of 8; 12.5%) [16], pilot studies (1 of 8; 12.5%) [21], and screening studies (1 of 8; 12.5%) [17]. These studies were conducted in four different countries: three in the United States (37.5%) [18,19,21], two in China (25.0%) [16,17], two in Germany (25.0%) [22,23], and one in Canada (12.5%) [20]. Four studies were enrolled on national (50.0%) [16,17,19,23] and four on local (50.0%) [18,20,21,22] levels, while none of the included studies were conducted at the regional level. In terms of the use cases, heart failure was represented in three (37.5%) [20,22,23], cardiovascular disease in three (37.5%) [18,19,21], and atrial fibrillation in two studies (25.0%) [16,17]. The smallest study cohort comprised 10 participants. The largest study included 246,541 participants. The minimum participation duration of all eight studies ranged from 14 days to 393 days. We provide an overview of the results in Table 2. 

With respect to the technical characteristics of the included studies, we identified six studies following an interventional monitoring approach (75.0%) [16,17,20,21,22,23], while two studies used the applied app and technology to log patients’ health status for further research (25.0%) [18,19]. The operating systems for the patient apps included Google Android and Apple iOS; the latter was used as a platform for conducting two studies (25.0%) [18,22]: one study was carried out using multiple platforms (12.5%) [19], and one relied on the use of an Android-based app (12.5%) [17]. The remaining four articles (50.0%) provided no further information about the platform(s) used [16,20,21,23]. When it comes to the wearables used, two articles stated the use of smartwatches from Apple (25.0%) [18,22], one article reported the use of a Fitbit wearable (12.5%) [21], one study relied on the use of a Withings smartwatch and Withings fitness tracker (12.5%) [19], while two articles reported the use of the Honor Band 4, the Honor Watch, and the Huawei Watch GT (25.0%) [16,17]. Two study protocols did not plan the use of any wearables (25.0%) [20,23]. Furthermore, we analyzed whether the participants were provided with other consumer health devices connected to the patients’ apps. We found that five study protocols included different types of Bluetooth blood pressure monitors (62.5%) [18,19,20,21], four involved the use of Bluetooth scales (50.0%) [19,20,21,23], and one study each included the use of a glucometer (12.5%) [21], a sleep tracking system (12.5%) [19], an electrocardiography device (12.5%) [23] and a pulse oximeter (12.5%) [23]. Some studies used a combination of several of the aforementioned devices. Three studies did not use additional devices aside from the wearables (37.5%) [16,17,22]. We provide an overview of the results in Table 3. 

Based on previously defined groups of PROMs, we could categorize two PROMs as outcomes measuring functional statuses (2 of 17; 11.76%) [18,23], five as describing health-related quality of life (29.41%) [20,22,23], three for symptoms and symptom burden (17.65%) [16,20], one for personal experience of care (5.88%) [20], four for health-related behaviors such as anxiety and depression (23.53%) [18,19,22], and two non-disjunct PROMs (11.76%) [18,22]. The exact allocation of the PROMs can be found in the (Appendix A: Categorization PROMs).

In Table 4, we provide an overview of the quantitative and qualitative parameters described within the reviewed studies. Overall, 17 PROMs could be identified, with two studies using one PROM (25.0%) [18,21], five studies using two or more types of PROMs (62.5%) [16,18,20,23], and no PROMs reported in one study (12.5%) [17].

When it comes to PGHD, five studies (62.5%) [16,17,18,21,22] collected various parameters, while three studies did not foresee the documentation of any additional data by the patient (37.5%) [19,20,23]. These five studies took into account a variety of self-documented lifestyle factors, such as diet (2 of 8; 25,0%) [16,21], smoking behavior (2 of 8; 25.0%) [18,21], and alcohol use (1 of 8; 12.5%) [18]. Furthermore, therapy compliance factors such as medication adherence (6 of 8; 75.0%) [16,17,18,21,22] were documented, while unspecified health surveys (1 of 8; 12.5%) [18], self-reported risk factors (1 of 8; 12.5%) [18], information about cardiovascular disease history (1 of 8; 12.5%) [18], sociodemographic data (1 of 8; 12.5%) [18], atrial-fibrillation-related hospital visits (1 of 8; 12.5%) [17], and hospitalizations (1 of 8; 12.5%) [17] were also requested to be entered into the patients’ app or paper-based questionnaire. One study asked the patients to enter their blood pressure and weight manually into the app (12.5%) [22], using non-connected conventional devices. In addition to the aforementioned patient-reported data, three studies reported the assessment of laboratory parameters at the beginning and in the course of the study (37.5%) [16,20,23]. One further study used a clinical questionnaire for the collection of data by clinical staff (12.5%) [17]. One study conducted a six-minute walk test and an ECG examination by study personnel (12.5%) [22]. Based on the wearables and devices to be found in Table 3, a wide range of self-tracked parameters could be identified, including seven studies measuring the patients’ heart rate (87.5%) [16,17,18,19,20,22,23], six studies measuring the patients’ blood pressure (75.0%) [16,18,19,20,21,23], four studies asking the patients to track their weight (50.0%) [20,21,23], and two using the devices to track the daily steps or mean daily steps (25.0%) [19,21]. Finally, the device-based self-tracking of a six-minute walk test (6MWT) [22], no further described physical activity [19], and the measurement of blood glucose [21], pulse wave velocity (PWV) [19], sleep duration [19], and oxygen saturation (SpO2) [23] were each performed in one study (12.5%).

Werhahn et al. equipped patients with the Apple Watches to measure their heart rate. They used built-in pedometer functions of both smartphones and Apple Watches to capture daily steps, calculated as an arithmetic mean of 14 days. During three planned study site visits, the device-based 6MWT was validated by simultaneously carrying out a regular 6MWT [22]. Wenger et al. report that their trial participants measured their blood glucose levels daily using a glucometer, as well as their daily steps using the Fitbit’s built-in pedometer; moreover, they collected participants’ blood pressure and bodyweight once a week on the same day using a Bluetooth BP monitor and weight scale [21]. Seto et al. did not use any wearables, but did use Bluetooth BP monitors and weight scales to measure heart rate, blood pressure, and bodyweight daily [20]. Modena et al. included patients already owning a Withings fitness tracker or Withings Watch and BP monitor, weight scale, or sleep-tracking system to track their participants’ pulse wave velocity, blood pressure, heart rate, and bodyweight at least two days a week, while the participants’ physical activity levels were captured using the built-in activity trackers on the participants’ smartphones. Additionally, Modena et al. described measuring the participants’ sleep duration via a Withings smartwatch or a sleep-tracking system if available [19]. McManus et al. report that they equipped a subpopulation of their study cohort with an Apple Watch and an additional Bluetooth BP cuff to log their blood pressure weekly as well as their daily measured heart rate [18]. Guo and Wang et al. included participants owning a Huawei Watch GT, Honor Watch, or Honor Band 4 to frequently capture their heart rate every 10 min [17]. Guo and Lane et al. used the same selection of devices to capture both heart rate and blood pressure, but did not provide further information about the frequency [16]. No other consumer health devices were used in either setting described by Guo et al. [16,17]. In contrast, Koehler et al. outline the application of only non-wearable-based sensors, including ECG monitors, BP measuring devices, weighing scales, and SpO2 sensors; these four devices were used to track the participants heart rate, blood pressure, weight, and capillary oxygen saturation daily [23]. 

## 4. Discussion

This systematic review summarizes the findings of studies using a patient app as an interface to document not only different sensor-based vital signs, but also self-tracked and self-documented real-world health data, for the purpose of telemonitoring in cardiology and observational research, including cardiological telemedicine data. The results suggest that different types of commercially available wearables and other consumer health devices can be implemented in a meaningful way in order to gain major insights into health behaviors and the course of diseases in different cardiological patient cohorts.

The comparison shows that although the studies’ primary focuses were different, there are many similarities, suggesting that the symbiosis of these new technologies in a cardiological context seems to be of interest to researchers worldwide. To achieve their respective objectives, all studies relied on a combination of apps and non-invasive devices. While the interventional studies’ approach was to monitor the daily management of disease progression or to provide active support, preventing deterioration when serious symptoms occurred, the observational programs aimed to provide further real-world health data for medical research, improving therapies and treatments in the long term.

Furthermore, the comparison shows that the choice of non-invasive devices is crucial when it comes to monitoring either high-frequency data or snapshots of a patient’s health status. This also depends on the scientific question or the context of treatment. In the studies reviewed, sensor- and app-based monitoring was implemented on the basis of various cardiological use cases, while some had intersections when it came to the PROMs or self-tracked follow up parameters collected. As vital signs such as heart rate and blood pressure or weight were taken into account by almost all of the studies reviewed, it can be assumed that these turn out to be key physiological signals to be monitored, providing initial insights into a patient’s general condition. However, this is countered by the fact that the accuracy of commercial wrist-worn devices is subject to ongoing scientific debate [25,26,27]. From a monitoring point of view, wearables have the advantage that they can provide high-frequency streaming data while worn. Although the market for consumer health devices is rapidly evolving, the types of sensors used in commercially available wearables are still limited, e.g., blood pressure, heart rate, SpO2, electrocardiogram, or photoplethysmography; thus, the need for both further developments in current sensors (e.g., wrist-worn ECG with more leads) and new sensor technologies was also recognized in the studies examined. This is why in some studies additional consumer health devices were applied to add follow-up parameters that generally cannot yet be captured by wearables or cannot be captured with sufficient quality. Adding to this, the review found that frequent surveys of standardized PROMs via a patient app seem to be another meaningful way to assess various aspects of a patient’s health status at home by adding further assessment criteria. Moreover, the digitization of PROMs seems to be a meaningful step towards a more patient-centered treatment [28,29]. While, from our point of view, for purposes of analysis, the use of structured data acquisition is to be preferred, there is much to be said for expanding the data basis through simple surveys, such as confirmation of medication intake or documentation of dietary behavior, as practiced in some of the programs.

Werhahn et al.’s study required patients to manually enter self-measured body weight as well as other parameters into the app without fully exploiting the possibilities of automatically transferring measurements by using existing interfaces such as Bluetooth. In contrast are Seto et al.’s, Moderna et al.’s, and Koehler et al.’s approaches to reduce the hurdle for regular data transfers to the app by equipping patients with Bluetooth scales. Thus, the manual entry of patients’ medical history by the patients themselves, as described by McManus et al., has potential for improvement, as this data could already be stored in the patient’s EHR or personal health record (PHR). Seto et al. describe a practical example, as they explicitly mention the import of laboratory parameters—e.g., brain natriuretic peptide (BNP) levels—from their hospital’s EHRs. Furthermore, Koehler et al. also took the BNP level into account, while Guo et al. took hemoglobin, liver, and renal function in both screened studies into consideration for the prediction of deterioration of the state of health. This review did not investigate whether or how laboratory parameters were transferred to the app but, again, it seems reasonable to do so by integrating the EHRs. Among all of the studies considered, Wenger et al. were the only team to use a point-of-care test, as synchronized glucometers to measure patients’ blood glucose were handed to the participants. This demonstrates that further laboratory parameters, which can currently only be measured by health care professionals, could in the future also be measured in the home setting. This would add a wider range of parameters to be monitored. The general advantages of mHealth technologies consist not only of bridging time and distance, but also offer the potential to avoid resource-intensive on-site monitoring. As soon as more over-the-counter sensors for measuring laboratory parameters reach market maturity, further scientific and clinical value could be gained from their integration in monitoring concepts. However, this is yet to be evaluated in further studies.

Finally, it is important to consider the platforms used, as the review revealed that only Modena et al. took a cross-platform approach integrating real-world health data from both Android and iOS devices. In the other studies, patients were provided with a compatible smartphone, or were only eligible for study participation if they already owned a suitable device. Consequently, this automatically leads to the exclusion of potential patients with unsupported device combinations. When considering a multiplatform approach, the corresponding effort and associated resource consumption must be taken into account. While a less complex single-platform approach allows the full exploitation of features of wearables or other devices via native interfaces, a comprehensive and elaborate integration into a multiplatform application might be associated with limited access to all device features [30]. Koehler et al., for example, integrated various consumer health devices from different manufacturers, although the underlying platform was unknown to the authors.

In summary, although consumer health devices or wearables remain evolving technologies, they are already able to offer a meaningful contribution in providing a more holistic insight into cardiological patients’ health status and behavior, while at the same time bridging the distance between patient and doctor.

### 4.1. Limitations

The results suggest that the search terms used were appropriate for the research question, but some limitations of our study should still be considered. For instance, our keywords telemedicine or telecardiology could limit the choice to studies that focused on interventional approaches, while observational studies are left out. To weaken the impact, we added the keyword mHealth to our queries. This did not provide more results, and was therefore dismissed. Furthermore, as the title/abstract filter was not applied constantly for the PubMed query, this results in a slightly larger pool of findings, which had a positive impact on the scope of our results.

In addition to the selected search terms, the challenge was to create a category scheme in which all included studies could be meaningfully presented to provide a holistic overview without excluding relevant factors. Therefore, the scheme was limited to categories that are relevant from the authors’ point of view. However, all information can be found in a table in the Appendix A. The separation between PGHD, PROMs, and clinical parameters was also discussed and assessed in detail between the authors to accomplish it as distinctly as possible; thus, we cannot ensure that everybody would evaluate this in the same manner. Although prominent studies such as the Apple Heart Study [5,31] were not included in the literature review, we assume that our analysis covered studies in the clinical context of telecardiology. However, this indicates that there may be other studies in the field that we did not include.

### 4.2. Outlook

In the context of this review, we did not address the algorithms used—for example, by Guo et al. and Seto et al. to predict AF and decompensation in HF, respectively. Although there are already internationally agreed treatment standards, there is still a lack of transparent and uniform diagnostic algorithms, as these are the subject of current research. It could be of interest to investigate which cardiological therapy guidelines or standards have been used to derive rules for algorithms, and what is the status quo in cardiologic algorithm research. Thinking beyond study situations, the possibilities of regular patients contributing their self-tracked health data into their EHRs are also of interest. In addition, as we advocate the establishment of platforms through which users can donate their wearable data for public research purposes without being tied to a specific purpose, corresponding concepts could be of interest for further research.

In future studies, it seems appropriate to replace the manual documentation of sensory data (e.g., weight by integrating consumer health Bluetooth scales). Given this, suitable solutions satisfying regulatory, technical, and medical requirements will be sought. As a second improvement, the adaption of further or different questionnaires should be investigated.

## 5. Conclusions

In this systematic review, we evaluated different approaches conducted by various researchers in the field of cardiological patient monitoring, which applied an integrated combination of app-based surveys, wearables, and other consumer health devices. Our review shows that, depending on the researcher’s motivation, a fusion of apps, PROMs, and non-invasive sensors can be orchestrated in a meaningful way, adding major contributions to monitoring concepts for both individual patients and larger cohorts. We suggest that different combinations of device-based vital-sign monitoring combined with patient-reported outcomes and the documentation of lifestyle factors can contribute further insights into patients’ disease progression, therapy compliance, and general health behavior patterns. In the medium-to-long term, disease prevention will most likely depend on consumer-health-device-based cardiovascular risk monitoring as a tool to follow patients up.

## Figures and Tables

**Figure 1 ijerph-18-09070-f001:**
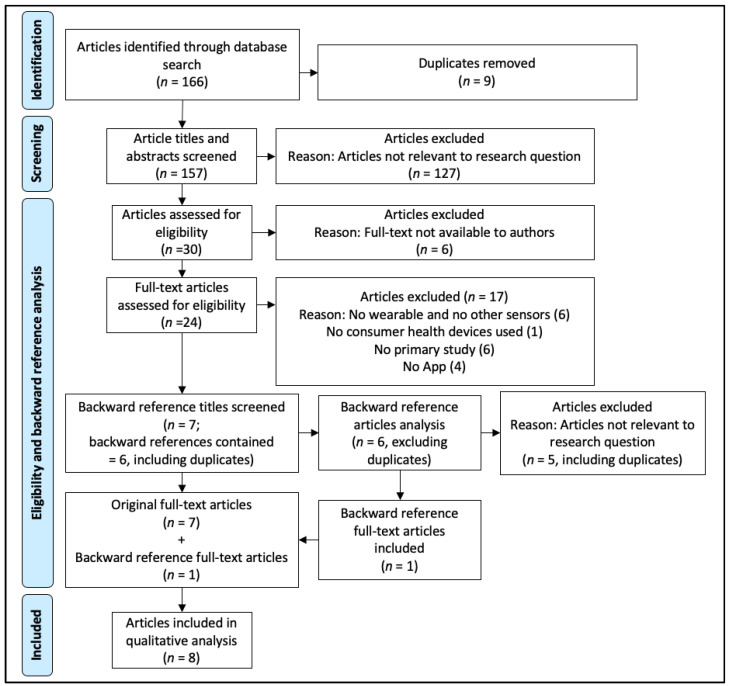
PRISMA flow diagram of the literature screening process.

**Table 1 ijerph-18-09070-t001:** Our search queries as we executed them per database.

Database	Query
*PubMed*	((telemedicine OR telecardiology [Title/Abstract]) AND (cardiology OR “cardiovascular disease” [Title/Abstract]) AND (app OR mobile application [Title/Abstract])) AND ((“2001/01/01” [Date—Publication]: “2021/03/31” [Date—Publication]))
*CINAHL*	((telemedicine OR telecardiology [Title/Abstract]) AND (cardiology OR “cardiovascular disease” [Title/Abstract]) AND (app OR mobile application [Title/Abstract])) AND ((“2001/01/01” [Date—Publication]: “2021/03/31” [Date—Publication]))
*Cochrane*	(telemedicine OR telecardiology):ti,ab AND (cardiology OR “cardiovascular disease”):ti,ab AND (app OR mobile application):ti,ab” with Cochrane Library publication date between Jan 2001 and Mar 2021
*Web of Science*	(AB = ((telemedicine OR telecardiology) AND (cardiology OR “cardiovascular disease”) AND (app OR mobile application))) OR (TI = ((telemedicine OR telecardiology) AND (cardiology OR “cardiovascular disease”) AND (app OR mobile application))) (Search period was set via the UI of *Web of Science*.)
*SCOPUS*	((ABS (telemedicine OR telecardiology) AND ABS (cardiology OR “cardiovascular disease”) AND ABS (app OR mobile AND application))) OR ((TITLE (telemedicine OR telecardiology) AND TITLE (cardiology OR “cardiovascular disease”) AND TITLE (app OR mobile AND application))) AND (LIMIT-TO (SRCTYPE, “j”) OR LIMIT-TO (SRCTYPE, “p”)) (Search period was set via the UI of *SCOPUS*.)

Proper names are shown in italics.

**Table 2 ijerph-18-09070-t002:** Overview of the studies included in the systematic review, with a focus on the study protocols.

Ref.	Country	Application Area	Study Type	Disease	Sample Size (Population Size)	Participation Duration
Werhahn et al., 2019 [22]	Germany	Local	Proof-of-concept study	Heart failure	10 (10)	2 months
Wenger et al., 2019 [21]	USA	Local	Pilot Study	Cardiovascular disease	14 (14)	6 months
Seto et al., 2020 [20]	Canada	Local	RCT ^1^	Heart failure	74 (144)	3 months
Modena et al., 2018 [19]	USA	National	Proof-of-concept study	Cardiovascular disease	250 (250)	17 weeks
McManus et al., 2019 [18]	USA	Local	Longitudinal cohort study	Cardiovascular disease	790 (4095)	≥3 months
Guo et al., 2019a [17]	China	National	Screening study	Atrial fibrillation	187,912 (246,541)	≥14 days
Guo et al., 2019 [16] [24]	China	National	CRT ^2^	Atrial fibrillation	32,259 (32,259)	≥14 days
Koehler et al., 2018 [23]	Germany	National	RCT ^1^	Heart failure	796 (1571)	365–393 days

^1^ RCT: randomized controlled trial; ^2^ CRT: cluster randomized controlled trial.

**Table 3 ijerph-18-09070-t003:** Overview of the studies included in the systematic review with a focus on the technical approaches.

Ref.	Monitoring (Interventional/Observing)	Operating System/ Platform	Wearable	Other Consumer Health Devices
Werhahn et al., 2019 [22]	Interventional	iOS versions 10.2.1–11.2.1	(1) Apple Watch 1st Gen.	/
Wenger et al., 2019 [21]	Interventional	Unknown	(1) Fitbit	(1) Weight-scale (2) Glucometer (3) Sphygmomanometer
Seto et al., 2020 [20]	Interventional	Unknown	/	(1) A&D Medical Bluetooth weight scales (2) A&D Medical Bluetooth BP ^1^ monitors
Modena et al., 2018 [19]	Observing	Android/iOS	(1) Withings fitness tracker (2) Withings Watch	(1) mHealth BP ^1^ monitor (2) Smart weight scale (3) Sleep-tracking system
McManus et al., 2019 [18]	Observing	iOS versions 9 or higher	(1) Apple Watch	(1) Nokia Withings Digital BP ^1^ cuff
Guo et al., 2019a [17]	Interventional	Android 5.0 or higher	(1) Honor Band 4 (2) Honor Watch (3) Huawei Watch GT	/
Guo et al., 2019 [16]	Interventional	Unknown	(1) Honor Band 4 (2) Honor Watch (3) Huawei Watch GT	/
Koehler et al., 2018 [23]	Interventional	Unknown	/	(1) Three-channel ECG device: PhysioMem PM 1000, GETEMED (2) A&D BP ^1^ measuring device (UA767PBT) (3) Seca 861 Weighing scales (4) SpO2 ^2^ Signal Masimo Extraction Technology

^1^ BP: blood pressure; ^2^ SpO2: oxygen saturation.

**Table 4 ijerph-18-09070-t004:** Overview of the studies included in the systematic review, with a focus on the qualitative and quantitative parameters.

Ref.	PROM	PGHD	Clinical Parameters and Scales	Self-Tracked Follow-Up Parameters	Frequency
Werhahn et al., 2019 [22]	(1) Minnesota Living with Heart Failure Questionnaire (MLHFQ) (2) Kansas City Cardiomyopathy Questionnaire (KCCQ) (3) Patient Health Questionnaire Depression Scale (PHQ-9) (4) Cardiac Anxiety Questionnaire (CAQ) (5) eHealth literacy (questionnaire similar to the eHealth Literacy Scale)	(1) Self-measured blood pressure (2) Self-measured body weight (before breakfast) (3) Confirmation of medication intake	(1) Holter electrocardiograms (ECGs) with 4 days of records (2) Six-minute walk test (6MWT)	(1) Mean daily step count (MDSC) (2) Heart rate (3) Six-minute walk test (6MWT)	(1), (2) Daily (3) Three times at site visits
Wenger et al., 2019 [21]	(1) Individualized questionnaires (e.g., for medication adherence in case of missing BP measurements)	(1) Taking insulin or oral diabetes medication (2) Cholesterol medication (3) Following a diabetic healthful diet (4) Smoking cessation	/	(1) Blood glucose (2) Blood pressure (3) Weight (4) Daily steps	(1) Daily (every morning) (2) (3) Weekly (on the same day) (4) Daily
Seto et al., 2020 [20]	(1) Self-Care of Heart Failure Index (SCHFI) (2) Kansas City Cardiomyopathy Questionnaire-12 (KCCQ-12) (3) 5-level EQ-5D (EQ-5D-5L) (4) Shortness of Breath Scale	/	(1) Routine blood test (creatinine, sodium and potassium levels) (2) Brain natriuretic peptide (BNP)	(1) Weight (2) Blood pressure (3) Heart rate	(1), (2), (3) Daily
Modena et al., 2018 [19]	(1) Perceived Stress Scale Survey	/	/	(1) Pulse wave velocity (PWV) (2) Physical activity level (3) Blood pressure (4) Heart rate (5) Sleep duration (6) Weight (BMI)	(1), (3), (4), (6) ≥2 days per week (2) Tracked using built-in activity trackers on the participants’ smartphone (5) Daily
McManus et al., 2019 [18]	(1) Center for Epidemiologic Studies Depression Scale, (CES-D) (2) Physical activity index (FHS)	(1) Socio-demographics (2) Medication use (3) Self-reported risk factors (4) Smoking (5) Alcohol use (6) Health survey (7) CVD history/non-CVD medical history	/	(1) Blood pressure (2) Heart rate	(1) 1 day per week at the same day (2) Daily
Guo et al., 2019a [17]	/	(1) Medicine usage (2) Visits for AF-related adverse outcomes (3) Hospitalizations	(1) HAS-BLED score ^1^ (2) Congestive heart failure, hypertension, age > 75, diabetes, stroke, vascular disease, age 65–74 years, and sex category (CHA2DS2-VASc) (3) Female sex, age, medical history, treatment, tobacco use, race score (SAMe-T2T2R)	(1) Heart rate	(1) Every 10 min
Guo et al., 2019 [16]	(1) Patient-reported thromboembolism or bleeding events (2) Atrial Fibrillation (AF) symptom assessment scale from the European Heart Rhythm Association (EHRA)	(1) Drug adherence (dose and drug use) (2) Patient-specific cost diary	(1) Hemoglobin, liver, renal function (2) HAS-BLED score ^1^	(1) Blood pressure (2) Heart rate	Unknown
Koehler et al., 2018 [23]	(1) Minnesota Living with Heart Failure Questionnaire (MLHFQ) (2) Self-rated health status (scale range 1–5)	/	(1) Follow Up Visit Biomarker (2) N-terminal prohormone brain natriuretic peptide (NT-proBNP) (3) Mid-regional proadrenomedullin (MR-proADM)	(1) Weight (2) Blood pressure (3) Heart rate (4) Heart rhythm peripheral capillary oxygen saturation (SpO2)	(1), (2), (3), (4) Daily

^1^ HAS-BLED: hypertension, abnormal renal/liver function, stroke, bleeding history or predisposition, labile international normalized ratio, elderly, drugs/alcohol concomitantly score.

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
