# Peer review of "Telemonitoring of Real-World Health Data in Cardiology: A Systematic Review"

_ijerph, 2021, doi:10.3390/ijerph18179070_

Round 1

Reviewer 1 Report

Manuscript revision:  ijerph-1339599.pdf                                                                                       11 August 2021

Title: Telemonitoring of Real-Health Data in Cardiology: A Systematic Review

Authors: Benjamin Kinast, Mathias Lutz, Börn Schreiweis

Keywords: telemonitoring, telemedicine; telecardiology; cardiology; wereable; sensors; consumer health devices; cardiovascular disease; heart failure; atrial fibrillation.

Dear authors, the work is very nice, and I think it is very well written.

The core of your work consists of the selection of 8 papers about the monitoring of patients with cardiology disease.

I think you should try to remark what is the novelty of your work, although it consists on a literature review. Nowadays there are many papers that address the monitoring of patients with cardiology disease.

Major reviewer recommendations:

1.- The manuscript objectives are well described at the end of Introduction section, but moreover, the novelty of the study should be mentioned or explained at the end of the Introduction.

2.- If the manuscript topic is studied for the first time in the geographical area, the interest and importance of the subject should be emphasized.

3.- The different sections that compose the manuscript should show briefly but with some detail at the end of the Introduction section.

4.- Section 2: it should be “Place, Data and Methods”. Some information similar to the information given on Page 3, paragraph 111-113 pages should be included in “Place”.

Minor reviewer recommendations:

1- Line 115: please correct “inlucding”

2- Line 163. Please, revise the spelling in this line.

Author Response

  1. We highly appreciate this suggestion and therefore added further information to the introduction pointing out the uniqueness of the conducted review:

    »With this novel approach we aim to provide a holistic perspective on telemonitoring as we take both the general organizational and technical context as well as qualitative and quantitative aspects into consideration.«
  2. As this article is a systematic review, the manuscript is considering a global context and not limited to a single geographical area. Furthermore, we assume that this review is not only of national, but of international interest. Of course, we would have liked to consider corresponding publications from other countries, but our search results are limited to the results presented.
  3. We are unsure whether we understand this comment correctly. We chose not to present the scientific and methodological structure of the article in the introduction, as this would contradict the standards known to us. A brief overview of the individual chapters of the article is found in the structured abstract.
  4. We agree that place is an important criterion. However, since the chosen heading "Materials and Methods" is a default of the journal, we choose to keep it. With reference to the second part of the comment, we decided to keep the chosen structure, since “Place” or “location of the study” in our opinion is only a single distinguishing criterion of the study protocol. Designating a separate subheading for this item would result in the paragraph below it being short in content and length. 

Reviewer 2 Report

An overview of telemonitoring of actual health data in cardiology is interesting and valuable. It should be emphasized that a systematic review of studies assessing the usefulness of new devices in monitoring cardiological parameters is very important and should be helpful in constructing a universal model for monitoring the most common cardiovascular diseases.

Congratulations to the authors, the article is very well written. 

The article is valuable because in the current literature there are very few papers comparing the use of modern technology in the daily monitoring of cardiological parameters. It would be interesting to discuss the possibility of using new devices for the assessment of more detailed biochemical parameters in the future. According to the authors, will the blood glucose level measurement with the new Apple Watch Series 8 be reliable? Will it be possible to perform cholesterol measurements using a similar technology in the near future? What other parameters do the authors propose?

Author Response

  1. This comment is very valuable to us as it addresses the need we discussed for more sensors for an even more comprehensive monitoring capability. At the same time, we are unfortunately unable to make any statements about the current validity of temperature or blood glucose sensors in an undisclosed Apple Watch 8, as these must first be certified by the FDA as medical devices before an investigation appears meaningful. In addition, current information about the Apple Watch 8 is based on patent leaks that still have to be confirmed by the manufacturer. The research results of Wenger et al. show that it is possible to integrate blood glucose measurements in a meaningful way therefore we consider the potential value of a wrist-worn blood glucose sensor to be very high, especially in terms of everyday usability and ease of use.

Reviewer 3 Report

Nice works

Author Response

  1. Thank you for your positive feedback. It is very much appreciated.

Round 2

Reviewer 1 Report

 I recomend the paper for publications